# Dynamic Interactive Model of Sport Motivation

**DOI:** 10.3390/ijerph19074202

**Published:** 2022-04-01

**Authors:** Matej Tušak, Donatella Di Corrado, Marinella Coco, Maks Tušak, Iztok Žilavec, Robert Masten

**Affiliations:** 1Department of Social and Humanistic Sciences in Sport, Faculty of Sport, University of Ljubljana, Gortanova 22, 1000 Ljubljana, Slovenia; 2Department of Sport Sciences, Kore University, Cittadella Universitaria, 94100 Enna, Italy; donatella.dicorrado@unikore.it; 3Department of Biomedical and Biotechnological Sciences, University of Catania, 95123 Catania, Italy; marinella.coco@gmail.com; 4Department for Medicine Science, Sirius AM, 1358 Log pri Brezovici, Slovenia; maks.tusak@gmail.com (M.T.); izilavec@gmail.com (I.Ž.); 5Department of Psychology, Faculty of Arts, University of Ljubljana, 1000 Ljubljana, Slovenia; robert.v.masten@gmail.com

**Keywords:** sport motivation, model of motivation, elite athletes, young talented athletes

## Abstract

Motivation variables in 11 motivational instruments of 357 Slovenian male athletes (168 elite and 189 young athletes from age 12–14) in nine different sport disciplines (basketball, football, handball, water polo, ice hockey, ski jumping, alpine skiing, sport climbing, and judo) were obtained. Different concepts of motivation were researched, such as achievement motivation, incentive motivation, participation motivation, goal orientation, satisfaction and enjoyment in sport, self-efficacy, effort, and ability attributions. The most popular framework for motivation in sport lately has been social cognitive perspective. The aim of this study was to form a dynamic interactive model of sport motivation. We tried to upgrade different models of motivation to one unique meta model of sport motivation, which would explain possible behaviours and motivation in sport situations. Different statistic methods were used to define differences among young and elite athletes and between athletes in group and individual sports. The results show important differences among those groups and suggest that specific sport discipline also has a specific footprint inside motivation. Factor analysis and discriminant analysis were used to explore sport motivation space. The results also suggest that it is possible to define some main determinants of sport motivation that can be connected to previous models of sport motivation.

## 1. Dynamic Interactive Model of Sport Motivation

Motivation for sport activities has been a very popular area in the field of sport psychology. Although a lot of researchers have tackled this area, the basic determinants of motivation for physical activities are yet to be found. Some very interesting problems have occurred because researchers did not separate the phenomena of level of involvement in sport quite exactly. Some of the researchers have researched top sports, others college sports, or other forms of fitness and recreation activities. Their approaches are mostly partial and just directed in investigating localized problems. However, motivation is a very wide construct. We are trying to see motivation as a very complex phenomenon, which must be researched freely with all its correlating variables.

Success in competitive sport depends mostly on an athlete’s skills, personality, and motivation. Many studies were conducted to investigate determinants of motivation. The presence of “zeitgeist” social cognitive perspective in psychology has changed the view on motivation for sport. Social cognitive approaches became the main framework for investigation of sport motivation. Social cognitive prospective started with the work of Weiner [1] and is built around expectancies and values that individuals attach to different goals and achievement activities. We can divide the social cognitive approach into three mini-theories:The theory of self-efficacy [2,3],The theory of perceived competence [4,5], andThe theory of goal perspectives [6,7,8,9,10,11].

Self-efficacy [3] is a common cognitive mechanism for mediating athlete’s motivation, thought patterns, and behavior. The self-efficacy construct has been used to explain achievement behavior in sport. Self-efficacy refers to one’s perceived ability for a particular task [2]. Different studies indicate that self-efficacy has a positive effect on performance in individual sports [12,13,14,15] and in muscular endurance tasks [16], but there is a question of the relation between self-efficacy and collective efficacy and collective performance in group sports.

Researchers have tried to explain why people want to participate in achievement situations [4,5]. A prediction of Harter’s model [4] is that children who perceive themselves competent in sport should be more likely to participate in sport activities. In addition, many others [16] have found similar results as previous studies [17,18,19,20], that this relationship is very weak and they suggest that there are many different reasons for children’s participation in sport.

Participation and persistence in sport and the choice and intensity of training and participating are goal directed. The goal is subjective, and the effect of multiplicity of different goals is presented in the process of motivation [21,22,23,24,25]. The success and failure in performance are not always defined according to winning or losing in the competition [26,27]. There are two major goal perspectives, or ways of defining success:

Task involvement or goal orientation: the focus is on learning, improvement, and meeting the demands of the activity: “trying to do athlete’s best”, “to perform perfect”, etc., to reach personal goals, where perceived competence is self-referenced and the subjective experience of personal improvement and task mastery defines success),Ego involvement or win orientation: the focus is on wining, “being the best” and showing the superiority over others is the primary goal; perceived competence is normatively referenced and depends on comparison of one’s ability to others.

According to many studies [11], the major goal of achievement behavior is to demonstrate ability and avoid the demonstration of low ability. The development of task and ego goals is a direct result of an emerging capacity to differentiate ability from effort as causal attribution of success and failure. Task goals are related to mastery, co-operation, sportsmanlike behavior, enjoyment, and the belief that effort leads to success in sport [27,28,29]. Ego goals are related to unsportsmanlike behavior, aggression, and the belief that high ability leads to success [30,31,32,33,34,35,36,37].

Socialization appears to be the strongest determining factor of athletes’ ego and task involvement. The parents and coaches become very important in building motivational climate, which directs athlete’s goal perspectives [38]. The sport setting is characterized by an increasing emphasis on competitive outcomes and normative ability as the athlete moves through the sport system (from junior to top athlete). Achievement orientation is a function of both development differences and situational constraints [25].

The participation motivation approach is focused on the reasons why people engage in sport and continue in their athletic participation [16]. Different researchers have found from five to eight primary goals or incentives for participating in sport. These are: health motives, achievement, team, friendship, fitness, energy release, skill development, and fun. Nicholls theoretical work [9,38] suggests that there is a link between goal orientation and participation motives. Dispositional goal perspective that an athlete brings to a particular situation will impact an athlete’s motivation.

Although achievement goal theory and the self-determination theory were published decades ago, they are still dominant theories in scientific explanation of a latent construct of motivation in sport [39]. Modern integrative hierarchical and multidimensional models of motivation in sport are based on these two basic theories [40].

In one of latest studies [41], on the basis of a systematic analysis of 63 studies on competitive sport motivation between 1995 and 2016, researchers discovered that that self-determination theory (SDT) and achievement goal theory (AGT) or integration of both represent a vast majority of theoretical backgrounds of studies. This was especially true for self-determination theory, which was in one study combined with Deci and Ryan’s cognitive evaluation theory (with emphasis on the role of social and environmental factors and their impact on intrinsic motivation) [41].

Further, evaluation of the six most highly cited questionnaires for measuring sport motivation confirmed that the intrinsic motivation inventory (IMI), which is based on self-determination theory, was most cited motivational questionnaire in 2016 with highest average weighted IF [42].

In this study we tried to form a dynamic interactive model of sport motivation. We tried to upgrade different models of motivation to one unique model, which would explain all possible behaviors and motivation in sport situation.

## 2. Material and Methods

### 2.1. Participants

The sample included 357 Slovenian male athletes altogether. Of this number, 168 athletes were between 17 and 30 years old (representatives of Slovenian national teams in basketball, football, handball, ice hockey, water polo, ski jumping, alpine skiing, sport climbing, and judo), and 189 boys were between 12 and 14 years of age—all of them young perspective athletes, who practiced and trained their sport in sport clubs for at least three years. Four main subsamples were defined: top athletes in individual sports (TI) (N = 80), top athletes in group sports (TG) (N = 88), young athletes in individual sports (YI) (N = 70), and young athletes in group sports (YG) (N = 119).

### 2.2. Measures

Measured motivational variables were as follows:Perceptions of demonstrated ability, effort, and self-efficacy [43]. All these constructs were measured on a 5-point Likert scale. There was one question regarding ability (test-retest r = 0.79): “What part of your accomplishment on the competition is the consequence of your ability?” and two questions for measuring effort: “How strong do you try on the competition?” and “How much effort do you put into the competition?” There were also two items for measuring self-efficacy: “How good do you think you compete in your sport?” and “How good are you in your sport discipline?” Cronbach alpha for effort α = 0.75 and for self-efficacy α = 0.77.Sport satisfaction and enjoyment [43] was obtained. Subjects had to evaluate their satisfaction with training, satisfaction with results, with participation and performance and with possibilities for training on a 5-point Likert scale. Cronbach alpha was α = 0.73.Expectations of results and success [43] (now, in the future, and in the entire career) were also measured on a 5-point Likert scale. Cronbach alpha was α = 0.70.Sport attitudes inventory [44] has been used to asses constructs related to sport behaviour and competitiveness on the competition. The first scale POWER motive measures the desire to have an impact on other people’s behaviour or feelings and include 12 items, with Cronbach alpha α = 0.70. Achievement motivation on the competition is represented by individual’s inclination to achieve success (MAS scale-motive to achieve success-positive competitive motivation, which includes 17 items with Cronbach alpha α = 0.72) and motivation to avoid failure (MAF scale-motive to avoid failure-negative competitive motivation with 11 items and Cronbach alpha α = 0.74. Dimensions are measured on a Likert scale from 1 to 5.Costello nAch questionnaire [45], which measures two achievement orientations on a Likert scale from 1 to 5: the need to achieve success with your own work and the need to achieve success regardless of your work. The first dimension includes 14 items with Cronbach alpha α = 0.69, and the second dimension includes 10 items with Cronbach alpha α = 0.82.Sport orientation questionnaire [46] was developed to assess the disposition to strive for success in competitive sport activities. The SOQ contains 25 items incorporating three subscales on a Likert scale from 1 to 5: Competitiveness (tendency to seek out or avoid the competitive situation) includes 13 items with Cronbach alpha α = 0.92, Win orientation (the desire to win in interpersonal competition in sport) includes 6 items with Cronbach alpha α = 0.86, and Goal orientation (the desire to reach personal goals in sport) includes 6 items with Cronbach alpha α = 0.82.Sport motivation scale [47] contains 50 items on a Likert scale from 1 to 5 and was developed to assess various motivational and personality dispositions in sport. It measures total score and five different sources and incentives of motivation for sport activities: aggression (α = 0.67), conflict (α = 0.80), competence (α = 0.72), competition (α = 0.70), and cooperation (α = 0.71). Each dimension contains 10 items.Self-motivation inventory [48] contains 40 items on Likert scale from 1 to 5. Cronbach alpha was α = 0.88. It measures self-discipline and self-motivation (e.g., When I start a heavy task I decide to persevere until I complete it).Task end ego orientation sport questionnaire [21] measures ego and task orientation on a 5-point Likert scale and contains 13 items. Ego orientation (7 items, α = 0.81) depends on an athlete’s perception of his abilities compared to others, success is a win or to be the best among all. Perception of success is based on social comparison. Task orientation (6 items, α = 0.89) is based on self-referred abilities; success is learning, improving performance, excellence in performance etc.Motives for competition scale [49] has a total score and 19 different subscales of incentives. All are measured on a Likert scale from 0 to 4 (very harmful for me to very useful for me). Scale contains 95 items; each subscale has 5 items. Cronbach α for total scale is α = 0.77.Participation motivation questionnaire [46] with the list of 30 motives for participation in sport and six factors latent structure: fitness and recreation motive, development of abilities, success and achievement, health, progression motive and challenge, experience of arousal and individuality, team atmosphere, and friendship. All motives were measured on a Likert 5-point scale.

On the base of these six factors, two new ones (of second order) were extracted: general participation motivation and specific participation motivation. For each factor, a factor score was calculated.

### 2.3. Procedures and Statistical Analysis

We invited 18 groups of athletes separately (nine sport disciplines, each in two age groups). Every group consisted of the best representatives of Slovenian athletes in each specific sport and separately by age. Subjects were requested to complete questionnaire items after the researcher had read the instructions. During application, researchers were present and available for potential questions. The athletes’ coaches were also present during application of instruments. Participants could use brakes to ensure their motivation on the appropriate level. Athletes were motivated to answer because they all received feedback about their motivation. Cronbach alpha coefficients and K-S test of normality were calculated. Analysis of variance was used for investigating differences between groups. Two factor analysis (PC) was calculated to define factor scores for the participation motivation questionnaire. Discriminant analysis was used to establish differentiation model of motivation, and factor analysis (PC) of all variables was used to set the model of motivation in sport. Statistical package SPSS was used for all statistical analysis.

## 3. Results

Discriminant analyses were made to discover the space of sport motivation (Table 1, Table 2, Table 3 and Table 4). Reduced set of the following variables were put into the analysis: ability and effort attribution of success (ABILITY), goal orientations (ego and task orientation, win and goal orientation) (EGO ORIENT., TASK ORIENT., WIN ORIENT., GOAL ORIENT.), competitiveness (COMPETITIV), nAch motivation (need to achieve success with work or no matter of work) (+nAch, −nAch), achievement motivation for competition (MAS, MAF, POWER motive), self-motivation (SELF-MOTIV), self-efficacy expectations (SELF-EFFICACY), success and result expectations (EXPECT.SUCCESS), general and specific participation motivation (GENERAL I., SPECIFI I.), total score of motivation for competition (TSMC), total score of motivation from five different sources on sport motivation scales (TSSMS), and sport satisfaction and enjoyment (ENJOYMENT, SAT).

Three discriminant functions were extracted (Table 1). They are all significant and form three dimensions of motivation that differentiate among four groups (elite athletes in group sport = EG, elite athletes in individual sport = EI, young athletes in group = YG, and young athletes in individual sports = YI).

Analysis of univariate differences (Table 2) shows the existence of important differences among all four groups of athletes (TI, TG, YI, and YG) in self-efficacy, win orientation, ego orientation, negative achievement motivation self-motivation, enjoyment, and specific factor of participation motivation and total scores of motivation for competition and in five subscales.

Discriminant function 1 (Table 3) includes motivation, which originates from incentive systems that are very attractive, important, and useful for athletes. It is their intensity and their power that are important for an athlete. These attractive motives stimulate athlete’s activities. Discriminant function 1 also includes negative nAch motivation and enjoyment in sport, but on the other side it indicates the absence of self-motivation and inherent control in motivation process. The first discriminant function indicates the “pull motivation”, like attractive incentive systems, the usefulness of motives for competition, feeling of some emotions, and expressing some personal dispositions. We named the first function the power of incentive motivation.

The second discriminant function (Table 3) includes general participation motivation (fitness and recreation motives, development of abilities, success and achievement, health, progression motives and challenge, team atmosphere, and friendship), ego orientation, and positive nAch motivation, but on the other side the absence of win orientation (which is related to group tasks and group directed goals and activities, such as cooperation) and competition. We named this function Ego motivation.

We found the most important correlations of third function (Table 3) with self-efficacy, total score of enjoyment in sport and specific motives for participation (motives to experience thrill, arousal, and individuality) and expectancies of success (in present and in the future). The function is negatively correlated with goal orientation and motives for power. The function was named Cognitive mediators of motivation.

Function the power of incentive motivation discriminates the most between young athletes in group (YG) and individual sports (YI) on one side (highly expressed) and top athletes in individual (TI) and group sports (TG) on the other side (less expressed) (Table 4). Function ego motivation discriminates between athletes in individual sports (TI and YI with high scores) and athletes in group sports (TG and YG with low score). It is quite difficult to find an explanation for the discrimination of function 3 (cognitive mediators). TI and YG reach higher results than YI and TG (Figure 1).

Factor analysis of all motivational space and variables extracted six determinants of motivation, which were described as intrinsic achievement motivation, self-regulatory mechanism and cognitive mediators, achievement orientation and personal characteristics, extrinsic achievement motivation, and two incentive systems (of general and specific incentive motivation). The first two factors are the strongest and together explain 38.2% of motivation (Table 5).

Table 6 shows exact saturations of the six factors with manifest variables. All factors together explain 62% of motivational space.

The most important motivational factor is saturated with self-motivation, +nAch (need to achieve success with hard work), power motive, motive to achieve success, effort attribution, and task orientation (Table 6).

## 4. Discussion

Analysis of motivational structure gave us a clear model. The first factor represents the most positive component of motivation in sport. The aim of such motivation is to achieve success. An athlete is aware that sport results depend on the athlete’s hard work and effort. Such an athlete is motivated by hard work (which he/she invests into the practice and competition), by progress, learning, and development of abilities. Such an athlete has strong intrinsic control and is self-motivated and goal oriented; he is also motivated by the possibility of influencing other participants in sport. This factor could be named intrinsic (positive) achievement motivation [1]. Results showed that this factor is the most important factor of motivation in sport, as it explains almost 30% variance. This intrinsic achievement motivation is also the most self-determined [50,51].

The second factor includes variables related to mediators of motivation. The role of self-efficacy as the mediator in the process of motivation was noted by many researchers [52,53,54,55]. A higher degree of self-efficacy leads to stronger goal setting and searching for more challenging goals that dictate stronger motivation. Even Bandura [2,3] located self-efficacy in the sphere of mediator inside his concept of cognitive motivation, which is goal oriented. A very similar approach was used for explaining motivation in sport by Dzewaltowski [56], with his concept of sport enjoyment and satisfaction in sport as one of the cognitive mediators of motivation. Satisfaction and enjoyment represent emotional self-evaluation, which is one component of self-regulatory influences [3]. Self-regulatory influences and experienced satisfaction in sport are important motivators in sport. Anticipating sport satisfaction and enjoyment (which go together with reaching athlete’s goals) have a strong impact on an athlete’s self-regulation [57]. Inside the concept of self-regulation constructs we can find also the attributes of success (ability and effort perception). Other researchers [9] already emphasized the self-concept of ability. Attributes of success represent the central mediator process in motivational situation. Cognitive representations of all those noted concepts of the second factor contribute to an athlete’s self-regulation process of motivation. This second factor represents Bandura’s [3] construct of self-efficacy expectations and cognitions related to self-reactive influences in the context of the process of self-regulation. We should not forget the expectation of success, which represents one of the three basic cognitive processes related to sport activities [2] and impacts an athlete’s perceptions of self-efficacy and competence [58]. This factor could be understood as self-regulatory skills, self-reactive influences, or cognitive mediators of motivation. Higher values on the second factor result in higher motivation behavior. High self-efficacy, clear expectations of results, and defined attributes of success lead to optimal cognitive motivation, which dictates endurance in training and sport behavior.

The third factor includes variables related to personal dispositions of achieving success. It represents an athlete’s achievement orientations in sport and training activities. We named the factor achievement orientations or personal characteristics of achievement behavior. It includes competitiveness, which discriminated between athletes and non-athletes [46,59], win orientation (includes tendency to win in interpersonal competition), and tendency to reach important personal goals through participation in sport (goal orientation). Inside the concept of social-cognitive perspective [2,56] we can find achievement orientations as personal determinants of sport activity. Achievement orientations are personal characteristics, but they are also affected by motivational climate, which represents the athlete’s social environment and the influences of the athlete’s process of socialization.

The structure of the fourth factor is quite unclear. It includes negative achievement motivation, ego orientation, and total scores of incentive systems. It suggests a kind of external and extrinsic achievement motivation. Externally motivated athletes are motivated with the fear of failure, they are ego oriented, and they do not care much about their own improvement and hard work. In the context of a self-determined continuum [50], such motivation lies somewhere on the lower level. The fifth and the sixth factors represent the attractiveness of incentive systems in sport. The fifth factor is called incentive systems of general motivation and includes attractiveness of all basic participation motives (achievement, recreation, skill development, group atmosphere, etc.), which motivate most of the athletes. The sixth factor is named incentive systems of specific motivation and includes the motive to experience thrill and excitement. Inside Banduras’ interactive model, both factors represent the incentive systems of the social environment, which could be understood as an athletes’ “pull” motivation [58].

The analysis of a scree test suggests a two or three factor solution. But a 6-factor solution was really interesting, and it would be very interesting to think about a suggestion offered by this model, that intrinsic and extrinsic motivation should be understood as two different dimensions of motivation and not just two ends of the same dimension. Results confirm that there is a possibility that an athlete expresses high or low scores on both dimensions at the same time. Very important in the present model is the dimension of cognitive process mediators, which touch on personal inclinations to different evaluations of success on the competition and different evaluations of the related result-goal-success. The attractiveness of incentive systems of the environment is suggested with the fifth and sixth factors and include general participation motives and specific motives that are very characteristic for extreme and high-risk sports (mountain climbing, ski-jumping, alpine skiing, etc.).

Comparison of this model with other models of motivation is possible. Combination of intrinsic achievement motivation, cognitive mediators, and specific motivation (first, second, and sixth factors) on one side and achievement orientations, extrinsic Ach. Motivation, and incentive system of general motivation on the other side suggests discrimination of motivation into intrinsic and extrinsic motivation [51].

If we consider just some of the factors in our model, we can also find even more similarities with older models of motivation. If we combine just our first and fourth factors, it can be compared with the concept of achievement motivation, but just the third factor can be compared with relatively narrow applications of achievement behavior in sport, so-called achievement goals perspective [9,10,60].

We can also find some similarities with the sport activity adopted attribution model [1], where the causal cognitions are represented by our second factor of cognitive motivation, goal expectations can be replaced with our incentive systems (fifth and sixth factors), behavior can be understood as achievement motivated (first and fourth factors of intrinsic and extrinsic Ach. motivation), and result can be evaluated with achievement orientations (third factor).

Our model can also be compared with the two-dimensional model of enjoyment in sport [61], which searches the source of motivation in factors achievement—no achievement motivation and intrinsic–extrinsic motivation. Our model perhaps leaves empty just one (of four possible) quadrant: extrinsic motivation in the no achievement context. The absence of extrinsic motivation in the no achievement context is probably the consequence of specific samples that included only athletes. The most interesting comparison of our model with a model of enjoyment is perhaps the greatest percent of variance explained by the factor of intrinsic achievement motivation. These results suggest that intrinsic satisfaction and intrinsic enjoyment could be understood as the main source of an athlete’s motivation.

The presence of cognitive motivation mediators confirms the importance of cognition in the process of motivation [62]. We have to understand our six factors inside the concept of a social-cognitive perspective, which sees motivation interactively associated in the factors of personality, environment, and behavior [3]. Personality includes athletes’ self-regulatory skills and cognitive mediators. Their goal orientation and incentive systems are influenced mostly by the environmental factors (motivation climate), but the behavior can be seen as achievement and no achievement oriented and on the other side as intrinsically and extrinsically motivated. Incentive systems (fifth and sixth factors) represent athlete’s goals, which, under self-regulatory skills (second factor) and goal orientation (third factor), effect achievement behavior (first and fourth factors), which was also explained by Bandura [3]. The stronger the athlete’s goal intentions and incentive systems are, the better and stronger his/her motivation will be.

The structure of the 6-factor model of motivation also includes the incentive motivation approach, but all six factors can also be interpreted inside the context of VE approaches, where cognitive mediators represent expectations, but incentive systems and goal orientations represent valence and value.

Many similarities can be also found if we compare our model with Roberts’ [17] dynamic model of motivation, but the differences come from his opinion about the main source of motivation (tendency to demonstrate high level of ability). His approach was importantly influenced by Nicholls’ [9] theoretical model of achievement orientation in sport. However, he only understood motivation as achievement oriented.

This study has some limitations that need to be noted. The proposed model is an attempt to define a metatheory of sport motivation that emphasizes only general determinants, but we know that sport motivation differs according to the type of sport, age, etc. Another limitation of the study is that it refers only to male athletes and does not include female athletes. Moreover, the model is currently valid only for Slovenia, and it is suggested to replicate it in other countries to evaluate the general validity of the model.

## 5. Conclusions

On the basis of the present results, a contemporary model of motivation in sport has been made. The dynamic interactive model of sport motivation is still not perfectly explained. We still see the importance to research the directions and relations between all dimensions and determinants of the model. The structure of the model should also be interpreted with the help of results from discriminant analysis, which confirm some of the differences between top and young athletes. The reasons for the differences could be found in different sport motivational climates inside top and young sport, so there is a question if we can talk about top and youth sport together or we have to consider them as two separated phenomena. On the other side, we should try to find a way to upgrade different models of motivation to one unique model that would explain all possible behaviors and motivation in sport situations. The present model (Figure 2) should be researched and evaluated inside a social-cognitive perspective, inside achievement motivation approach, and inside interactive dynamic process of all motivational determinants in the future.

## Figures and Tables

**Figure 1 ijerph-19-04202-f001:**
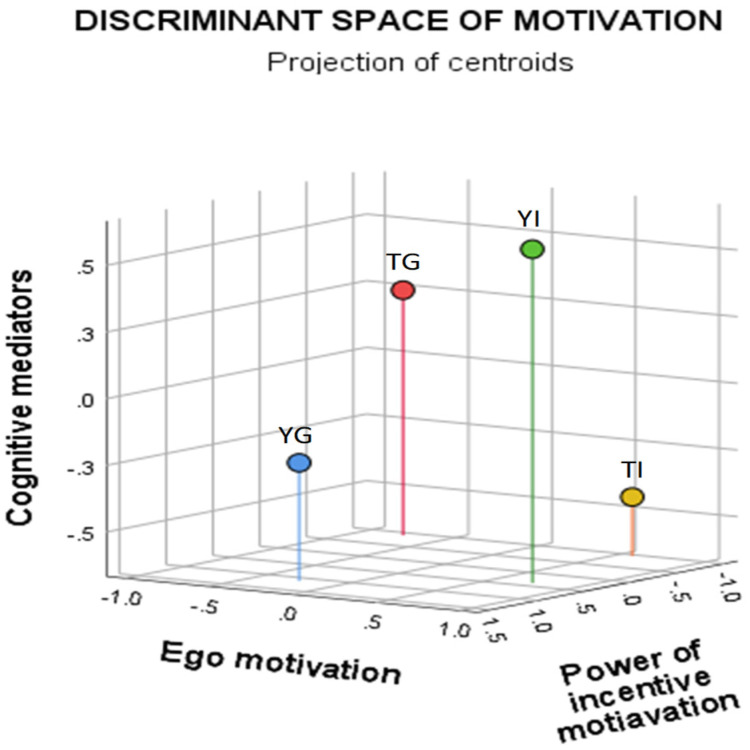
Projection of centroids for groups in 3-dimensional discriminant space of motivation.

**Figure 2 ijerph-19-04202-f002:**
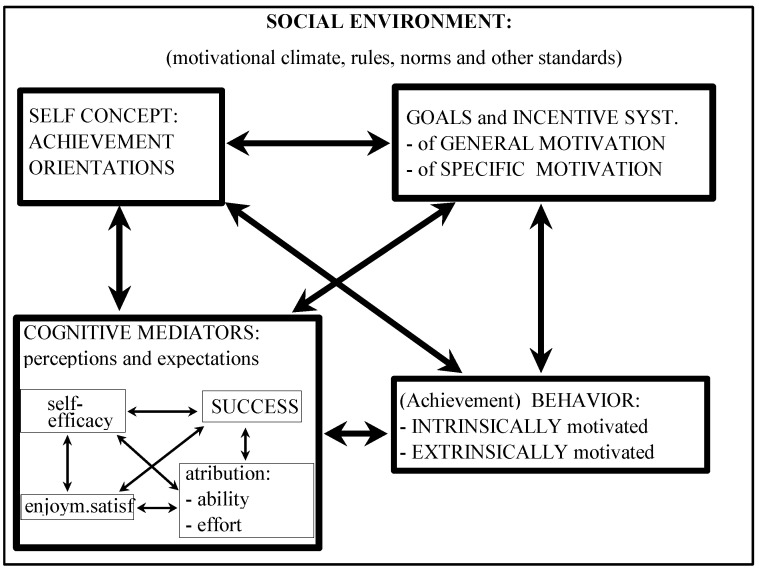
Dynamic interactive model of sport motivation.

**Table 1 ijerph-19-04202-t001:** Canonical discriminant functions.

Eigen Value	Pct. Var.	Cum. Pct.	Canon. Korr.	Fkc	Wilks’ Lambda	Hi-sq.	dF	Sig.
1.5214	57.69	57.69	0.7768	0	0.166964	553.10	162	0.000 *
0.7842	29.74	87.43	0.6630	1	0.420976	267.34	106	0.000 *
0.3313	12.57	100.00	0.4989	2	0.751124	88.43	52	0.001 *

*****—statistically significant at 0.05 level.

**Table 2 ijerph-19-04202-t002:** Wilks’ Lambda and univariate F.

Variable	Wilks’ Lambda	F	Fp.
ABILITY	0.995	0.534	0.659
EFFORT	0.999	1.138	0.334
SELF-EFFICACY	0.940	7.098	0.000 *
COMPETETIV.	0.979	2.350	0.072
WIN ORIENT.	0.923	9.267	0.000 *
GOAL ORIENT.	0.982	2.042	0.108
EGO ORIENT.	0.935	7.743	0.000 *
TASK ORIENT.	0.988	1.291	0.277
+nAch	0.995	0.569	0.635
−nAch	0.979	2.333	0.074
POWER	0.990	1.085	0.356
MAS	0.992	0.902	0.440
MAF	0.955	5.234	0.001 *
SELF-MOTIV.	0.953	5.485	0.001 *
ENJOYMENT, SAT.	0.814	25.545	0.000 *
EXPECT.SUCCESS	0.994	0.634	0.593
GENERAL I.	0.981	2.135	0.096
SPECIFIC I.	0.920	9.747	0.000 *
TSMC	0.883	14.824	0.000 *
TSSPS	0.856	18.764	0.000 *

Names of variables explained in text. *****—statistically significant at 0.05 level.

**Table 3 ijerph-19-04202-t003:** Structure matrix.

	Function 1	Function 2	Function 3
ENJOYMENT, SAT.	0.48677	−0.10544	0.38422
TSSPS	0.39672	−0.19330	−0.36987
TSMC	0.39014	0.07771	−0.02267
MAF	0.21718	−0.09374	−0.15156
SELF-MOTIV.	−0.20962	−0.17793	0.05677
TASK ORIENT.	0.11085	0.03913	0.05779
MAS	0.09082	0.01218	0.08044
WIN ORIENT.	−0.01330	−0.46297	−0.24388
EGO ORIENT.	0.07343	0.43211	0.03300
GENERAL I.	−0.24869	0.31456	−0.01018
COMPETITIV.	0.02047	−0.23558	−0.1009
+nAch	−0.02167	0.11644	−0.00808
POWER	−0.07769	−0.11193	−0.03731
ABILITY	0.03800	0.09608	−0.04903
SELF-EFFICACY	0.14825	−0.18373	0.47440
GOAL ORIENT.	−0.02043	0.09010	−0.32096
SPECIFIC I.	−0.07897	−0.01931	0.30481
-nAch	0.11212	−0.01492	0.26141
EFFORT	−0.08817	−0.04528	0.13845
EXPECT.SUCCESS	0.06401	0.04887	0.09845

Names of variables explained in text.

**Table 4 ijerph-19-04202-t004:** Group centroids and canonical discriminant functions.

Group	Function 1	Function 2	Function 3
YG	1.09825	−0.26010	−0.22630
YI	0.35040	0.71670	0.58227
TG	−0.92509	−0.77231	0.24975
TI	−0.97494	0.57201	−0.44771

YG = young athletes in group sports; YI = young athletes in individual sports; TG = top athletes in group sports; TI = top athletes in individual sports.

**Table 5 ijerph-19-04202-t005:** Factor analysis of reduced set of motivation variables (PC analysis, varimax rotation).

Factor	Eigen Value	% of Var.	Cum. % of Var.
Intrinsic achievement motivation	5.48	27.4	27.4
Self-regulatory mechanism, cognitive mediators of motivation	2.15	10.7	38.2
Achiev. orientation, personal characteristics of ach. Behavior	1.42	7.1	45.2
Extrinsic achievement motivation	1.27	6.3	51.6
Incentive system of general motivation	1.08	5.4	57.0
Incentive system of specific motivation (ind.m. and thrill exp.)	1.01	5.1	62.0

**Table 6 ijerph-19-04202-t006:** Saturation of factors with manifest motivation variables (only correlation coefficients > 0.40).

	Factor 1	Factor 2	Factor 3	Factor 4	Factor 5	Factor 6
SELF-MOTIV.	0.77					
+nAch	0.73					
POWER	0.61					
MAS	0.59					
EFFORT	0.55	0.42				
TASK ORIENT.	0.49				0.43	
SELF-EFFICACY		0.81				
ENJOYM., SAT.		0.79				
EXP.SUCCESS		0.67				
ABILITY		0.58				
WIN ORIENT.			0.80			
COMPETITIV.			0.79			
GOAL ORIENT.			0.77			
TSSMS				0.68		
MAF				0.68		
−nAch				0.66		
EGO ORIENT.				0.52		
TSMC				0.38		
GENERAL I					0.83	
SPECIFIC I.						0.85

Names of variables explained in text.

## Data Availability

The data presented in this study are not available.

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
