# Peer review of "Dynamic Interactive Model of Sport Motivation"

_ijerph, 2022, doi:10.3390/ijerph19074202_

Round 1
Reviewer 1 Report
This manuscript entitled "Dynamic Interactive Model of Sport Motivation" aimed to "form a dynamic interactive model of sport motivation"
This topic is quite relevant, but some importante issues require a resubmission, as follow:
- Please, put efforts to improve the abstract, which bring nothing about method, result and conclusion.
- All text and references must be updated. None study from the last 7 years are cited. What recent literature is telling about this topic? Which are the recent studies?
Author Response
First, thank you very much for your comments and your effort to make our article better.
Comment: Please, put efforts to improve the abstract, which bring nothing about method, result and conclusion.
Answer: The abstract was corrected acording to your suggestions
Comment: All text and references must be updated. None study from the last 7 years are cited. What recent literature is telling about this topic? Which are the recent studies?
We added a section related to the question of more recent studies. We analyzed some of latest studies and we confirmed, that although there were some specific sub-theories developed in general two basic theories remain actual (SDT and AGT). We added that at the end of “introduction“ section.
Reviewer 2 Report
Dynamic Interactive Model of Sport Motivation
First of all, the reviewer would like to thank the authors for their work and efforts in trying to improve sports science knowledge.
General comments to the authors
Overall, this is a nice study that could have great practical application. The authors are commended on their efforts thus far. The study is well designed and well-written, with a great original article evaluating the usefulness of the topic. However, I suggest only small corrections for manuscript.
Abstract
This section is well designed and well-written.
Introduction section
This section is well designed and well-written.
Methods section
This section is well designed and well-written.
Page 4 Line 151: which statistical package programme was used by authors?
Results section
This section is well designed and well-showed.
Discussion section
Overall the discussion is well-written and incorporates relevant literature.
Author Response
Information about statistical package was added on page 4 line 151. It is seen in added corrected documents.

Reviewer 3 Report
Thank you for allowing me to review this interesting study. Overall, the study has raised a very interesting point of discussion. I believe that this study has provided novel findings in this area, allowing readers to think more deeply about what is happening around physical activity. First of all, I would like to share the need to carry out works like the one you present. They are necessary for the advancement of science in the field they study. The purpose of the manuscript is clear and consistent. The study has been an interesting read, it is necessary to know the reality of the sector on which the work emphasizes.
In the introduction, sufficient ordered references of the publications considered key, with significant and sufficient evidence, are indicated. Likewise, reasons that justify the importance in a broad context and the current state of the investigated topic are highlighted. The study is clearly defined and indicates the intent and meaning of the work. The objective to be tested in the study is recorded. The text is understandable and makes clear the main objective of the work and the main conclusions. In relation to the material and methods, say that the study is described in detail. In addition to the methods, the intervention requirements are indicated in sufficient detail. Even so, it is considered that as a first contact to know the state of the matter is sufficient. However, the limitations of the work should be reflected. Not only do different age groups come together, but also different types of athletes. It should be considered that the motivation of team sports is not the same as in individual sports. As well as comparing athletes from the national team with young talents. I think it should be justified very well so that this can be representative of your country and can be replicated in another context. I would like to know the items that have been analyzed with the athletes, I would like to receive the annex of the instrument.
Author Response
In this study, we used standardised psychological instruments translated into Slovenian and adapted to the Slovenian cultural background. Therefore, we have only items in Slovenian language and we decided that items in Slovenian language are not interesting for readers. Also, some instruments could be restricted for a wider audience to prevent their use by unauthorised users.
We also described limitation of the study which were added on page 10, lines 326 till 331.

Reviewer 4 Report
Congratulations on your work, I think it can have great significance in the scientific world, although I wanted to comment on some considerations:
Some more data must be indicated in the abstract (such as the sample, the results and the "conclusions" in a summarized way)
In the introduction, I would like to see the first paragraph with substantiated bibliographical references. Likewise when talking about sporting success in the second paragraph.
In the method, I would like the cronbach's alpha of the questionnaires and the subdimensions to be included.
Reference 44 must be explained and the rest of the questionnaires as well (a little more detail about them). In the procedure and statistical analysis, indicate more exactly how it was (software, homogeneity tests, if atypical cases are detected...). Was the researcher present passing out the questionnaires? Were they face-to-face or online? (provide also an example of the questionnaire).
The questionnaire is very extensive. Do you think this could have influenced the answers? In turn, the data collection procedure should be further detailed.
The tables of the results should be explained, not just put them.
The discussion seems to be "a breakdown of the results", the first paragraphs should be eliminated and detailed in results (up to the figure, which should be adequately described according to the regulations of the journal)
Limitations section must be included.
Finally, please check the numbering carefully (for example, the number 41 in bold)
Author Response
Comment: Congratulations on your work, I think it can have great significance in the scientific world, although I wanted to comment on some considerations:
Answer: Thank you!
Comment: Some more data must be indicated in the abstract (such as the sample, the results and the "conclusions" in a summarized way)
Answer: The abstract was corrected acording to your suggestions
Comment: In the introduction, I would like to see the first paragraph with substantiated bibliographical references. Likewise when talking about sporting success in the second paragraph.
Answer: The introduction was changed quite a lot, we added some new references which help to understand the new findings in the area of sport motivation (at the end of introduction). Analisys of new researches was made.
Comment: In the method, I would like the cronbach's alpha of the questionnaires and the subdimensions to be included.
Answer: A lot of statistical data were added to method section, we added information about Cronbach alpha, K-S test, more explanation of the instruments were defined to better understand the researched sport motivation. For the instrument “Motives for participation“ alpha can not be calculated since the scores are made in factor analysis.
Comment: Reference 44 must be explained and the rest of the questionnaires as well (a little more detail about them). In the procedure and statistical analysis, indicate more exactly how it was (software, homogeneity tests, if atypical cases are detected...). Was the researcher present passing out the questionnaires? Were they face-to-face or online? (provide also an example of the questionnaire).
Answer: All answers were added to clarify your questions. Reference 44 is clarify inside the definition of instruments. First review were also made in the beggining and we found some atypical data (most of them they were found because of unprepareness and bad motivation of some participants to answer all instruments, so some data were blank and atypical: for example the subject scored all items five). They were just few of them, but we excluded these cases and the cronbach alphas then become appropriate.
Comment: The questionnaire is very extensive. Do you think this could have influenced the answers? In turn, the data collection procedure should be further detailed.
Answer: We answered these questions inside chapter procedures.
Comment: The tables of the results should be explained, not just put them.The discussion seems to be "a breakdown of the results", the first paragraphs should be eliminated and detailed in results (up to the figure, which should be adequately described according to the regulations of the journal)
Answer: We removed some text from discussion to results after tables, which adequately describe these tables and reorganise chapter results and we added some aditional description.
Comment: Limitations section must be included.
Answer: Limitations are added at the end of discussion.
Comment: Finally, please check the numbering carefully (for example, the number 41 in bold)
Answer: We corrected it.

Round 2
Reviewer 1 Report
All my suggestions were addressed by the authors.
Reviewer 4 Report
Dear authors.
I consider you have done a great job.
Congratulations.